# IndiLegalOnt: An Ontology for the Indian Legal System

Apurv Dube[1,*], Joseph Pookkatt[2], Raghava Mutharaju[1] and Vikram Goyal[1]

[1]*IIIT-Delhi, India*
[2]*Advocate, Supreme Court, India*

## Abstract

Legal tasks require managing complex information that is primarily unstructured in nature. There is a definite need for a more structured knowledge representation that can guide both human and machine learning models when building systems that aid in processing legal matters. An ontology provides a mechanism to build cohesive, structured knowledge which can be utilized for end-tasks via multiple mechanisms. We present *IndiLegalOnt*, an ontology that captures some of the aspects of the Indian legal system. We focus on reusing existing ontologies and limit the scope of this work to Indian court structures and selected regulations of the negotiable instruments act. We populate the ontology using our custom built library and evaluate it using the competency questions gathered from the domain experts. The IndiLegalOnt is available at https://github.com/kracr/indian-legal-ontology.

## Keywords

Indian Legal Ontology, Legal Ontology, Legal domain, SALI

## 1. Introduction

Law firms require tools to gather, analyze and share information between clients, vendors and other businesses. A consistent vocabulary is beneficial and a common format helps in avoiding translating information between multiple formats. Furthermore, to automate any task and aid in decision making, the legal firms need the ability to represent information in ways which are sufficiently expressive to model the complex entities and relations described in the laws. An ontology can be very useful here. It helps in providing a clearly defined vocabulary of terms, is helpful in exchanging information across applications and can represent facts and rules in a form that can be easily queried. When we combine an ontology with other techniques, this can help in tasks such as, documents classification, case information retrieval, and question answering.

Our work is motivated by a need to address the requirements of a legal firm, attempting to adopt a legal ontology for their work. We considered the existing ontologies in the legal domain and built an ontology by reusing them and incorporating additions that reflect the requirements of the law firm. Here, we focus on two such requirements. The first involved creating a hierarchical representation of the court system in India. Data about the Indian courts up to the district level were gathered and added to the ontology. Additionally, we reused an existing location ontology to bring robust location handling for India specific location data at the state, city and district levels. In the second set of additions, we looked at specific regulations in the Indian context relating to the Negotiable Instruments Act (NIA). The requirement was to model a few legal scenarios involving bad check crimes. There is a huge backlog of court cases in India in this category. Having an ontological representation would be useful in designing applications that can help the courts in speeding up their decision making process. Relevant classes and information regarding claims, defenses and rules of evidence related to the bad check crimes were modelled using the ontology. A team of lawyers and application developers were always in the loop. They provided the requirements, validated the design and provided feedback. We followed an iterative approach in designing, building, populating, and evaluating the ontology. The *IndiLegalOnt* is available at https://github.com/kracr/indian-legal-ontology.

---

*WOP 2024: 15th Workshop on Ontology Design and Patterns, Nov 12, 2024, Baltimore, Massachusetts, United States*
*Corresponding author.

✉ apurvd@iiitd.ac.in (A. Dube); joseph@apjlaw.com (J. Pookkatt); raghava.mutharaju@iiitd.ac.in (R. Mutharaju); vikram@iiitd.ac.in (V. Goyal)

## 2. Related work

### 2.1. Legal ontologies

Ontology engineering in the legal domain has gained significant traction over the past few decades, becoming essential for applications such as legal information retrieval, automated legal reasoning, and compliance management. Legal ontologies are pivotal in representing domain-independent legal concepts, properties, and relations, facilitating interoperability and standardization in legal knowledge representation. This section provides an overview of notable legal ontologies that have been developed to address various aspects of the legal domain, highlighting their unique contributions and applications.

Several core ontologies have emerged to standardize legal knowledge representation across different systems. The FOLaw[1] ontology, developed by Valente and Breuker, offers a functional approach to legal knowledge, focusing on the purposes and effects of legal norms. The Core Legal Ontology (CLO)[2], proposed by Gangemi, integrates legal theories with ontological principles to represent the structural and semantic aspects of legal texts. NM-L[3], developed by Shaheed, Yip, and Cunningham, bridges natural language thematic analysis with a commonsense view of reality, enhancing the interpretation of legal documents. The LRI-Core[4] ontology by Breuker and Hoekstra captures core legal concepts across various legal systems, emphasizing common-sense knowledge and practical reasoning. The LKIF-Core ontology[5], developed as part of the Estrella project, includes fundamental legal concepts such as legal agents, actions, norms, and roles to facilitate knowledge exchange. The LegalRuleML-Core[6] ontology, under the OASIS LegalRuleML initiative, focuses on the formal representation of legal norms and rules, supporting the interchange and automation of legal reasoning. LOTED2[7], created by Distinto, d'Aquin, and Motta, represents European public procurement notices, facilitating compliance and transparency in governmental contracts.

Other significant ontologies include the Legal Requirements Upper Ontology (LRUO)[8] by Breaux and Powers, which defines legal concepts for organizational compliance. The Top Ontology of the Law[9] by Hage and Verheij presents a dynamic view of legal states as interconnected events and rules. CausatiOnt[10], developed by Lehmann, Breuker, and Brouwer, models physical and agent causation within legal contexts, addressing issues of legal responsibility. Zarri's HClass[11] ontology represents narratives as events, capturing the complexities of legal narratives. The Legislative Ontology[12] by Costilla, Palacios, Cremades, and Vila, designed for the 'SIAP' parliamentary management system, which exemplifies the application of ontologies in e-government. Lastly, the SALI[13] initiative's Legal Matter Specification Standard (LMSS) provides a globally adopted taxonomy to enhance legal data interoperability and analytics, bridging substantive law with the business of law.

### 2.2. Rationale for Selecting An Ontology To Reuse

These ontologies represent a diverse array of frameworks developed to address specific needs and challenges in the legal domain. Each ontology contributes to the structured representation of legal knowledge, enhancing legal information retrieval, compliance, and decision-making processes. We further made a more detailed comparison of two potential candidates for our task, SALI and LKIF. The criteria for considering them was expressiveness, ease of use, development and adoption status. Despite its strengths in knowledge representation and reasoning, LKIF comes with a level of complexity which, while suitable for research and academia, can be challenging for businesses to interpret and implement. SALI, on the other hand, is significantly simpler and built for the business of law. This is the reason we proceeded with SALI. However, SALI is designed primarily for the US justice system. While it is broad enough to fit the requirements of any legal system, it needs specific extensions to it suitable for our requirements. For instance, government bodies, the court system, regulations and methodology of handling certain criminal cases, all differ to some extent between the Indian and the US legal systems.

# 3. IndiLegalOnt Description

## 3.1. Methodology

Legal experts, application developers, and ontology engineers worked together iteratively to build the ontology. Figure 1 illustrates this process.

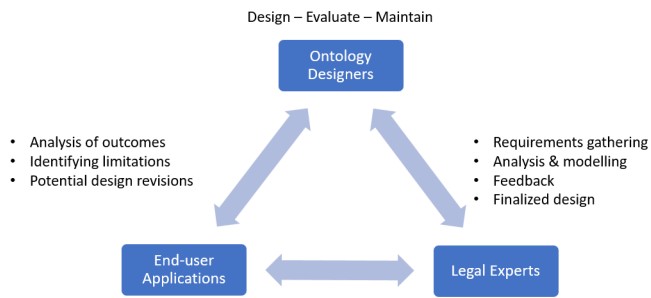

**Figure 1:** Design Process for ontology extension: Note that the legal experts were also part of the end-users.

We studied the structure of SALI ontology to isolate areas which needed to be adapted or extended to work for the Indian legal system. We consulted with legal experts regarding their requirements for representation and downstream tasks. Two primary aspects were identified as initial starting points to make meaningful additions.

The first was the addition of Indian entities such as government bodies, courts and tribunals. The Indian court structure differs significantly from the US system, and this was a useful starting point for our extension work. It also provides a glimpse into the interaction of courts with other entities.

In the second stage, we looked at regulatory gaps and how to represent case information relating to certain acts in India, such as the Negotiable Instruments Act (NIA). We chose this use case due to the volume of NIA cases that created a bottleneck in the Indian legal system. The goal here was to handle the bad-check crime cases which come under NIA and cause a backlog in the system. We noticed a marked difference in how bad-check crime cases were litigated and the claims and defences associated with them in the US versus the Indian legal system.

### 3.1.1. Competency Questions

Competency questions in ontology engineering are natural language questions that define the scope of knowledge an ontology should cover. They are a natural step after gathering requirements. They serve as functional requirements, ensuring the ontology can answer these questions. Some of the competency questions are provided here as an illustration.

- List all district courts in a given state.
- In which state is a given district court instance located?
- How many districts are in a given state of India?
- Which court has authority over a given court?
- What is the check transaction type for a bad check?
- When is a check payment of a bad check?
- What is the transaction status of a given check payment?

## 3.2. Ontology Design

### 3.2.1. Courts and Locations

The addition of Indian Courts was the first step in extending SALI. This was a huge task as information up to the district court level was added. To create a meaningful representation of court hierarchies and

their locations, we also incorporated a better way to represent locations within SALI. We chose a more standard approach and incorporated the GeoNames[14] ontology to represent the location of Indian districts, cities and states. Using a general purpose global ontology for locations allows us to extend it as required for places worldwide.

When it comes to specifying general axioms, the complex nature of some of the entities makes it harder to specify global rules. For instance, a case in a district court can only go up the chain via the high court of that state and finally to the apex court. Since our courts are closely tied to locations, it was easier to link instances with relevant properties. Similar issues arise when we try to have axioms defining the boundaries between cities, districts and states as illustrated in Figure 2 as the rules governing them are not consistent.

### 3.2.2. Classes, Properties and Axioms

We reused the SALI and GeoNames ontologes to model the court structure. Additions were made beneath the GeoNames Geographical region class and added subclasses pertaining to Country, State, District and City. GeoNames instance data IRIs for actual locations were later linked below these. We added the Indian Court structure below the Dispute Venue class of SALI. Properties connecting courts to show authority were also created and axioms representing precedence were created. For instance, all high courts in India are preceded by the Supreme court of India. Also, all courts in a state are preceded by the High court of the state.

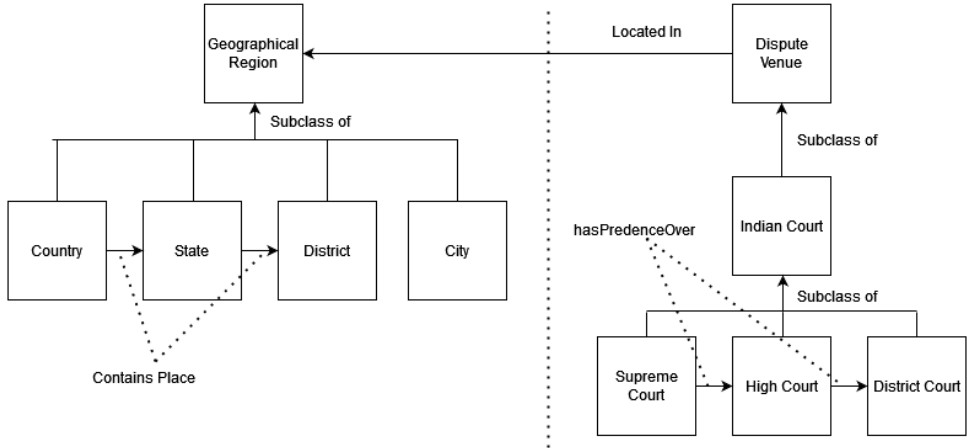

**Figure 2:** Addition of Indian court structure to SALI and incorporation of location using GeoNames ontology.

### 3.2.3. NIA related additions for bad-check crime

According to a 2023 report[15], check dishonour cases in India add up to over 8 percent of the overall criminal cases that are pending. This figure was estimated to be a staggering 351,600 cases! Our goal here was to construct the basic structure for managing information relating to such cases which come under the Negotiable Instruments Act in India. This could be used in tagging case files and prior judgements. Combining the ontology with other tools would potentially allow leveraging this data in determining what the outcomes were for similar cases. This provides the ability to aid in human decision making process based on a fully explainable line of reasoning.

We added classes to represent bad check transactions and the associated claims, defenses, and rules of evidence for bad check crimes. etc. There is a marked difference between how these cases are treated in the US system versus the Indian one. In the US, bad checks can be treated as civil and criminal offenses with varying state-specific penalties, whereas in India, issuing a bad check is primarily a criminal offense under the centralized Negotiable Instruments Act with strict penalties and mandatory notice periods.

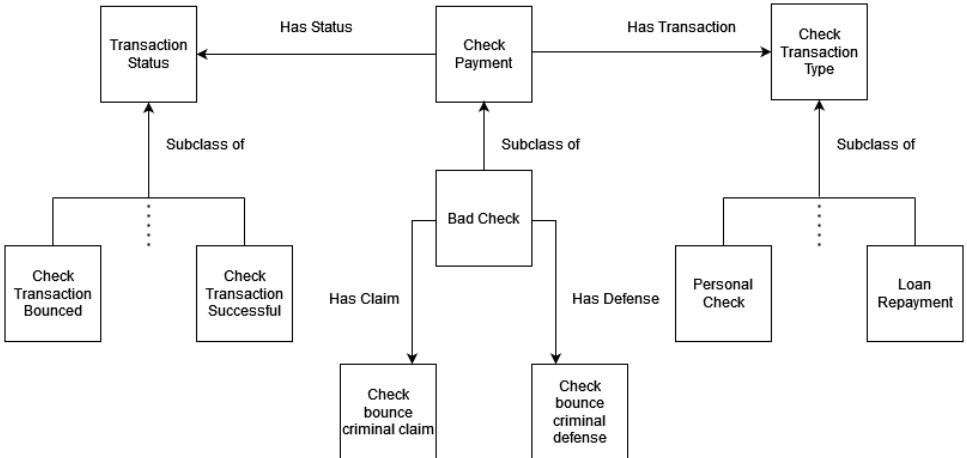

**Figure 3:** Modeling of bad-check crime cases. Note that this is a partial representation showing major classes.

### 3.2.4. Classes, properties and axioms

We have reused existing SALI classes relevant to the matter and figure 3 shows part of the classes added for managing bad check crime cases. In this scenario, we are modeling the types of check transactions and a bad check. Also, we link bad checks to classes representing criminal claims and defenses that allow one to capture a detailed description of the nature of the case. Classes like the transaction status, transaction type, criminal claim and defense are information dense and hence not shown here in their entirety. Axioms such as what defines a bad check were added to the ontology, i.e. A bad check is a type of check payment which has transaction status as check transaction bounced.

## 3.3. Populating the Ontology

Information about actual locations is stored as instance data in GeoNames and as expected the sheer size makes it difficult to be distributed along with the main ontology. GeoNames provides a web-API service which can be queried to extract relevant information. We queried this service to collect IRIs corresponding to the locations of interest and their names. This involved extracting information from GeoNames for administrative divisions at level 1 which correspond to states and union territories, respectively and level 2 which contain districts and then similarly for some select cities.

Once we created the structure of the courts, we included individual courts at each level as instance data. Data about the Supreme Court of India, 25 high courts and roughly 688 district courts was added. These instances were connected with relevant properties to represent the precedence of courts and their locations.

**Bulk additions to IndiLegalOnt.**   The two extensions required creating a tool to facilitate bulk addition. Java code using OWL API was written to make these current and future additions. This was necessary as manual additions beyond a few dozen via visual tools such as Protege became impractical and slow. Extensive Java code was written for the general library functions and the current set of additions.

The library offers a comprehensive suite of functionalities for ontology management. It allows users to create a new ontology or utilize an existing one, perform searches for subclasses or super-classes of a given class, and find classes by their labels. It supports bulk additions of new subclass hierarchies and individuals. Users can set entity types, add object and data properties, assert domain and range for these properties, and add labels and annotation properties, streamlining ontology development and maintenance. This tool is available at https://github.com/kracr/indian-legal-ontology.

## 4. Conclusion and Future Work

While we began with a few key additions, we understand that for any wider industry adoption, there are many more additions necessary for the Indian legal system. The volume and complexity of these additions make it a long-term project with iterative refinements over time. Some of these additions that we were modeling involved compliance handling, a broader framework for managing regulations and so on. With this initial ontology, we want to ascertain the utility of ontology in helping the legal industry in a multitude of downstream tasks. These include ways in which they tag case information, gather information from clients, analyze legal documents, and perform decision-making tasks. While some potential test cases were considered, the process is currently an ongoing challenge. One of the present tasks involved combining case data via RAG [16] based techniques to help direct Large Language Models (LLMs) while working with legal texts. The idea here is to prevent LLMs from hallucinating by incorporating the knowledge graphs for grounding them in factual data. Furthermore, knowledge graphs can also facilitate the generation of explainable results, which are invaluable in such domains. Given the nature of additions, we also are currently exploring semi-automate ontology modeling from text, using machine learning models, including LLMs. Such an approach could be useful in identifying additions or modifications and aid human-experts who would make the final decisions regarding such changes.

## Acknowledgments

The authors acknowledge the support of the Samagata Foundation, the Infosys Centre for Artificial Intelligence (CAI), IIIT-Delhi, and a team of lawyers and applications developers at Kronicle Research Private Ltd.

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
