# OpenReview forum: "IndiLegalOnt: An Ontology for the Indian Legal System"
_swsa.semanticweb.org/ISWC/2024/Workshop/WOP — WOP 2024 Oral_

### Official Review · Reviewer_iJSH · 2024-08-27
**The paper was easy to go through, figures were clear and competency questions enhanced the overall use case understanding. The paper could benefit from further discussion of the limitations of the approach when it comes to using ontologies from different legal systems. Also, the aspect of reproducability should be visited more thoroughly, especially when the use case targets legal acts.**

**Rating:** 6
**Confidence:** 3

**Review:**

# Introduction

**Pros**


Explains why the ontology is necessary by pointing out the distinctions between
the legal systems of India and the United States. gives an argument for the
need for this ontology and explains how it closes a gap that other ontologies
have not addressed. This lays a strong basis for the work's significance and
applicability.

**Cons**


Some aspects of the Indian legal system, like court systems and the Negotiable
Instruments Act, are the main focus of the research. It allows extent, but it
can also restrict the ontology's relevance to other legal fields, which could
reduce its usefulness for more general legal applications.
The introduction could be made easier to understand by providing a brief
overview of the importance of ontologies in the legal field.
A more balanced assessment of the work could be made by examining into the
specific challenges or constraints faced throughout the ontology's creation,
rather than just mentioning the iterative method.

# Related Work

**Pros**


SALI and LKIF are fairly compared when significant factors like expressiveness,
usability, and adoption rate are taken into account. This provides an argument
which assists in the decision-making process and helps to validate the
selection of SALI for the project.

**Cons**


While they state that SALI comes primarily from the US justice system, it seems
the further elaboration of how complex the adaptation would be for the Indian
system.
A further context given regarding legal ontologies would be apprechiated from a
readers perspective.
Further explanation as to how their work differentiates with other approaches
seems to be lacking.


# Methodology

**Pros**


An incremental contribution.
The legal experts' and researchers' cooperative approach improves the
ontology's usability and technical soundness.

**Cons**
Although the approach highlights the importance of filling in regulatory gaps
specific to the Indian legal system, it doesn't go into great depth about how
these gaps were systematically found and filled.

# Conclusions and Future Work

**Pros**


They recognize that integrating the ontology into real-world applications such
as case tagging and decision-making continues to bring difficulties.

**Cons**
However, they do not offer neither a clear path forward nor any concrete
solutions to these problems.
Looking into LLMs for automation is a great next step, how would the accuracy
be addressed though?

---

### Official Review · Reviewer_f8z5 · 2024-08-27
**The authors formally introduce IndiLegalOnt which tackles representation of Indian Legal System by adopting and extending existing ontologies.**

**Rating:** 4
**Confidence:** 4

**Review:**

While a Github repository is available and licensed, the repository itself lacks comprehensive documentation. The authors highlight 7 CQs as "some" of the competency questions to ensure the scope and quality of IndiLegalOnt is justifable; however, the repository only contains queried results of CQs #1, 2, and 3 (as listed in the publication). Additional explanation to their selected CQs would benefit the publication as to how these CQs individually gauge the quality of the ontology.  This could include how their CQs bridge together external datasets to enhance querying performances or explanations to why these 7 CQs are essential to the evaluation of the ontology.

As illustrated in Section 2, there already exists ontologies in the legal domain; however, each country still uniquely enforces their own laws and regulations. The authors poll together these existing technologies to build IndiLegalOnt as a first iteration of ontologically representing the Indian Legal System. A quick cursory search found that other researchers have made attempts specifically to tackle representing the Indian Legal System, however, from differing angles [1]. It is understandable why existing formal definitions and ontologies require tuning to any particular country's; however, the unanswered question would be why the authors opted to creating a newly defined ontology rather than implementing their methodology to expand on existing KGs (like NyOn).

Minor comments:
- Citation 14 contains unknown question mark characters in one of the bibliography fields.
- Sections 3.2.2 and 3.2.4 share similar titles

References
[1] Kanoon Sarathi - A Multilingual Portal for the Indian Judicial System. https://ihub-anubhuti-iiitd.org/blogpisarika.html.

Strengths:
- Github available and licensed
- Adopts existing ontologies and makes adjustments to better represent India's legal system

Weaknesses:
- Not a novel creation, as other KGs exist for representing India's Legal System
- Writing suggests other CQs were posed but cannot be found.
- Results of check-based CQs are missing
- Lack of detailed and comphrensible documentation found from Github repository
- Figures are modeled from partial representation; however, there are no full representation of the ontology's schema diagram.

---

### Official Review · Reviewer_yu48 · 2024-09-06
**Review for IndiLegalOnt**

**Rating:** 6
**Confidence:** 5

**Review:**

This paper presents additions to the SALI and GeoNames ontologies for the purposes of representing the Indian Legal System.

It provides some competency questions, discusses the additions made to the ontologies, and code to populate the ontology. This is clearly relevant to the workshop (i.e., ontology design). It would have been more interesting to also see a discussion of possible patterns for inclusion into the work.

There are some minor complaints: the figures could more clearly delineate additions vs. what was in SALI/GeoNames. What is the dashed line mean in Figure 2? bad-check is inconsistently hyphenated. Is there a proper legal term for this in the Indian Legal literature? At least by the first sentence of Section 3.2.3, it should be `check dishonour` perhaps this should be used more consistently instead.
precedence vs authority.

I generally have a favorable review of the paper, and think that discussions of the complexity -- especially with considerations of the reuse criteria will be interesting for the workshop.